# Osteoma of the Jaw as First Clinical Sign of Gardner’s Syndrome: The Experience of Two Italian Centers and Review

**DOI:** 10.3390/jcm12041496

**Published:** 2023-02-14

**Authors:** Silvia D’Agostino, Fabio Dell’Olio, Angela Tempesta, Francesca Cervinara, Antonio D’Amati, Marco Dolci, Gianfranco Favia, Saverio Capodiferro, Luisa Limongelli

**Affiliations:** 1Department of Medical, Oral, and Biotechnological Sciences, University G. d’Annunzio, 66100 Chieti, Italy; 2Complex Operating Unit of Odontostomatology, Department of Interdisciplinary Medicine, Aldo Moro University, 70121 Bari, Italy; 3Operating Unit of Pathological Anatomy, Department of Emergency and Organ Transplantation, Aldo Moro University, 70121 Bari, Italy

**Keywords:** Gardner’s syndrome, osteomas, familial adenomatous polyposis, desmoid tumors, supernumerary teeth

## Abstract

Gardner’s syndrome (GS) is a combination of polyposis, osteomas, fibromas, and sebaceous cysts. The aim of the study is to highlight whether maxillofacial osteoma could represent an early detection symptom of GS. Patients with suspected osteoma of the jaw underwent genetic and radiographical examinations. The database gathered 19 patients with oral osteoma that was histologically diagnosed; the whole sample was positive for APC gene mutation. Other cranial and peripheral locations were reported. Osteoma of the jaw is a crucial predictive factor of GS, and dentists and oral and maxillofacial surgeons must be aware of the importance of a timely diagnosis.

## 1. Introduction

Gardner’s syndrome (GS) is a multiple-systemic disease [1], counting for 10% of all familial adenomatous polyposis (FAP) patients. GS is a rare autosomal dominant disease, with complete penetrance for polyposis, variable penetrance for extra-intestinal features, and variable expressivity, affecting both genders equally. GS is characterized by the triad of FAP, multiple osteomas, and soft tissue tumors, but only 38% of all patients show the classic triad [2]; most patients present only one or two typical characteristics. Diagnosis must be confirmed by gastrointestinal endoscopy and molecular genetic analysis [3,4]. Depending on the author, GS’s incidence is 1 person per million, 1 case out of 8300 people in the United Kingdom, and 1 out of 12,000, 1 out of 14,025, or 1 out of 16,000 in Europe [3,5,6,7].

This syndrome is caused by mutations in the Adenomatous Polyposis Coli (APC) gene coding for the 300 kDa Anaphase Promoting Complex protein, on chromosome 5 in the q21-22 region [3,8]. A defective gene is inherited from one of the parents in approximately 75% of cases. Spontaneous mutations occur in 25–33% of cases [3,4,5]. More than 300 different mutations have been reported [3]. The variable clinical phenotypes depend on where the mutation occurs; the APC gene regulates intestinal tissue development. APC is a tumor suppressor gene and its mutations have been correlated to epithelial tumors: GS patients inherit a mutation and, following the “two hits” model, a subsequent additional somatic mutation results in the loss of heterozygosis and tumor development [6,7,8].

GS is a premalignant disease because its polyps show dysplasia (median diagnosis age 40 years) and an extremely high risk of malignant transformation (near 100% without treatment; generally, between 58% and 100%), which occurs between 30 and 50 years of age, with a mean age of diagnosis of 39.2 years, generally 10–15 years after the onset of the lesions. Colonic carcinoma occurs from the second to seventh decade; approximately 25% of patients with FAP have colorectal cancer (CRC) at diagnosis and die by the age of 50 years unless they receive surgical treatment; only rare reports describe patients free of malignancy [3,4,5].

Multiple osteomas occur in between 62% and 82% of GS cases; they are commonly asymptomatic but grow, gradually reaching a considerable size and causing disfigurements such as facial dysmorphism [9,10]. GS osteomas occur with two distinct patterns of distribution: focal (80%) or widespread (20%). Patients with more than three osteomas need to have a family history check for intestinal diseases because this is highly suggestive of GS: almost 50% of all FAP patients show at least three osteomas [2,3,4,5,6].

Dental abnormalities in the GS population are present in 30–75% of cases. GS patients present from 4% to 38% more impacted teeth and from 11% to 27% more supernumerary teeth than in the overall population [7].

Soft tissue lesions develop in 10% of GS patients [7] and include epidermoid cysts in 50–65% of patients, located commonly on the face, forehead, scalp, back, neck, and/or extremities.

GS has also ophthalmological manifestations, especially congenital hypertrophy of the retinal pigmented epithelium (CHRPE), which is present in up to 80% of cases and appears in the fundus shortly after birth, thereby resulting as a strong identifier rather than the first marker of GS [2,5,6,10,11].

Diagnosis of GS may involve the family history, presenting clinical features, intestinal examination, and genetic testing. The aim of this study is to report on the oral hard tissue lesions detected in 19 patients diagnosed at the University Hospital of Bari and the University Dental Clinic of Chieti from 2018 to 2021, highlighting whether maxillofacial osteomas could represent an early detection symptom of GS. A literature review is also provided.

## 2. Materials and Methods

This study was carried out following the principles of the Declaration of Helsinki and approved by the Independent Ethical Committee active at the University A. Moro of Bari, Italy (Study Protocol 1528/CE).

### 2.1. Inclusion Criteria

Patients were screened if they met the following inclusion criteria: single or multiple osteomas suspected in the upper/lower jaw, to be confirmed by histological analysis; osteomas of the jaw histologically diagnosed from 2018 to 2021; positivity on APC gene mutation test.

### 2.2. Study Design

Patients positive for the inclusion criteria provided informed consent for diagnostic and therapeutic procedures and possible use of their biological samples for research purposes. As a first step, general and specialistic anamnesis was achieved from patients or derived from medical records, including CRC family history. After the WHO eight-step intraoral examination, the following clinical data were collected: age, gender, intra- and extraoral osteoma locations and numbers; presence/absence of craniofacial pain, presence/absence of facial asymmetry, presence/absence of intestinal polyposis (or FAP), presence/absence of CRC and family history of FAP and CRC, presence/absence of soft tissue lesions, referred symptoms. Panoramic radiographs were performed in all instances associated with CBCT if necessary. To obtain a certain diagnosis, bone samples were collected from supposed osteomas and underwent histological analysis. Patients received customized clinical and radiological follow-up. Patients with a known FAP/CRC family or personal history were included in a recall program for clinical and radiological evaluations, whereas those with a negative history for both FAP/CRC received APC gene testing and further gastrointestinal investigation. The study methodology is shown in Figure 1.

### 2.3. Literature Review

A systematic review was conducted using the Preferred Reporting Items for Systematic Reviews and Meta-Analyses (PRISMA) guidelines for systematic reviews and meta-analyses [12]. To identify relevant studies investigating the presence of oral osteomas before GS diagnosis, a comprehensive search of the PubMed, Scopus, and Web of Sciences (WOS) databases, using the Patient/Population/Problem, Intervention, Comparison, and Outcome (PICO) format, was conducted.

Population: Adults, adolescents, and children.Intervention: Osteomas of the jaw in GS patients.Comparator: Systemically healthy subjects.Outcome: The earlier presence of oral osteomas.

The following MeSH terms were used: oral osteomas AND Gardner’s syndrome. No time or language restrictions were applied. The eligibility criteria were as follows: all studies analyzing the composition of the presence of osteomas in the oral and maxillofacial district in GS patients, focusing on their early onset. The exclusion criteria were as follows: cases having other systemic disorders, animal studies, and other predictive signs.

## 3. Results

### 3.1. Study Results

From 2018 to 2021, our database gathered 19 patients with oral osteomas. Every patent was positive for APC gene mutation. The cluster presented a mean age of 47 ± 17.08 years, with 52.6% of men (10 patients) and 47.4% of women (9 patients), and the average age for women was 42 ± 17.68 years old, while that for men was 52 ± 15.86 years old. A total of 24 oral osteomas were found, of which 66.7% were in the lower jaw (16 lesions) and 33.3% in the upper jaw (8 lesions). Moreover, 26.3% (5 patients) showed other cranial locations, including the frontal sinus and the occipital, the parietal, and the ethmoid bones, with a total of six cranial lesions. Meanwhile, 36.8% (7 patients) reported peripheral osteomas, such as in the clavicle, humerus, rib, ulna, tibia, and fibula. Most patients reported no craniofacial pain, and only one case was barely symptomatic because of the osteoma’s proximity to the mental foramen. Moreover, 31.6% (6 patients) revealed facial asymmetry. With regard to intestinal involvement, 68.4% (13 patients) had polyposis, while only 5.2% (1 patient) developed CRC. Concerning soft tissue tumors, 21.1% (4 patients) denoted dermoid cysts, whereas only 10.5% (2 patients) displayed epidermoid cysts. No dental abnormalities were found. The main extraoral GS patients’ features are summarized in Figure 2, whereas Figure 3 and Figure 4 show clinical cases chosen from the study sample.

### 3.2. Literature Review Results

The initial website research provided a total of 186 articles. In detail, there were 82 articles from PubMed, 47 from Scopus, and 57 from WOS. Excluding ineligible articles and duplicates, 17 items were suitable and focused on the PICO question. The PRISMA results are shown in Figure 5. To the best of the authors’ knowledge, a brief narrative description of the included items is supplied in Table A1. All studies reported multiple osteomas as the main early sign of GS, whereas dental abnormalities and soft tissue tumors were early signs of minor importance.

## 4. Discussion

An observational study was carried out in order to anticipate GS diagnosis in people responding to specific inclusion criteria. The study was conducted by considering the cases gathered in two Italian tertiary referral hospitals during the study period from 2018 to 2021. The current patients showed multiple osteomas as the main early signs of GS, and these data were in line with the results of the review of the literature. Only 31.6% of patients showed soft tissue tumors such as dermoid cysts and epidermoid cysts, and the latter result fits with the findings of the review. No dental anomalies were found in the study patients, in contrast with the literature review, and this finding could be due to the small size of the current study sample. The variable clinical phenotypes of GS depend on where the APC gene mutation occurs. Osteomas occur within the mutation spectrum spanning codon 767 to codon 1513; patients with upstream or downstream mutations do not develop osteomas [6]. The APC gene regulates intestinal tissue development, apical-basal polarity, the cell cycle, and DNA replication and repair, as well as apoptosis and the negative regulation of the Wnt/β-catenin tumorigenic pathway, which induces β-catenin accumulation in the cytoplasm, affecting proliferation, differentiation, migration, and apoptosis. GS’s adenomatous colonic polyps develop frequently in the colon and rectum, even if they may occur in any region of the gastrointestinal tract except the esophagus; gastric polyps are rare [3,4,5]. The standard diagnosis of typical FAP is based on the identification of >100 colorectal adenomatous polyps [2]. The average age at GS diagnosis ranges between 13 and 31 years, while common symptoms related to colonic polyps include intermittent mucous discharge with defecation, rectal bleeding, diarrhea, constipation, and abdominal pain. As stated by Antohi et al. [13], the extracolonic manifestation of GS could be divided into non-malignant and malignant. Regarding the oral and maxilla facial regions, prevalent signs are represented by osteoma, dental abnormalities, CHRPE, benign cutaneous lesions, desmoid tumors, and adrenal masses. Osteomas’ mean linear growth rate was estimated to be 0.117 mm/year (95% CI, 0.004–0.230 mm/year) [9,10]. GS’s osteomas occur with two distinct patterns of distribution: focal (80%) or widespread (20%). Their most common locations are at the angle of the mandible (bilaterally), maxilla, frontal bone, skull, paranasal sinuses, long bones, orbit, [1], and condylar process. These bone lesions can be classified into central osteomas, arising from the endosteum (enostoses), appearing as large and diffuse lobulate cotton wool-like lesions in the jaws, and peripheral osteomas, arising from the periosteum (exostoses), radiographically round or oval radiopaque masses attached by a broad base that may show also a uniform sclerotic pattern or periphery with a central trabecular pattern, while they clinically arise as polypoidal or sessile masses from the surface of the bone [10]. Histopathological examination distinguishes compact osteomas (also called “ivory” osteomas), composed of hard dense bone with minimal marrow spaces and occasional Haversian canals, and spongious osteomas (also known as mature osteomas), containing bone trabeculae and fibrofatty marrow with osteoblasts [9]. Patients with more than three osteomas need to have a family history check for intestinal diseases because this is highly suggestive of GS: almost 50% of all FAP patients show at least three osteomas [2,3,4,5,6]. Osteomas can be incidentally found on routine panoramic radiograms; thereby, dentists play a crucial role in GS diagnosis; CBCT provides better characterization than panoramic radiograms, which may underestimate the exact number of osteomas.

Dental abnormalities in the GS population are present in 30–75% of cases. Unerupted, congenitally missing, impacted, or supernumerary teeth, hypercementosis [2], ankylosis [7], odontogenic keratocysts, dentigerous cysts [6], fibromixoma [7], unicystic ameloblastoma and ameloblastic carcinoma [11], and compound odontomas [5,6,7] are the most common situations. Supernumerary teeth are present in 11–27% of the FAP population, compared with 0–4% of the general population.

Other tumors may appear simultaneously with osteomas, such as sebaceous cysts, thyroid carcinoma, periampullary carcinoma, osteosarcoma, lipoma, liposarcoma, hepatoblastoma, adrenal adenoma and carcinoma, leiomyoma, neurofibroma, fibroma stained with anti-CD34 and anti-β-catenin, (intra-parotid) pleomorphic adenoma, locally invasive desmoid tumors, pilomatrixoma, and pigmented skin lesions; the latter can be located also intraperitoneal or retroperitoneally; the female-to-male ratio for soft tissue lesions is 3:1 [3,4,5,8,9,10,11]. Both osteomas and soft tissue lesions are early manifestations of GS as they usually develop during puberty and may occur up to 10 years earlier than the formation of FAP [3,4,5].

Due to the high incidence of osteomas in FAP patients and GS patients, a particular focus should be reserved for this benign tumor. Oral osteomas are found in 60–80% of FAP patients and 50% of all oral osteomas occur in FAP. This incidence is remarkable, especially if compared to the normal occurrence of osteomas in the healthy population, which is approximately 1–2% [13]. In particular, Capodiferro et al., 2005 [14] suggested that dentomaxillofacial lesions in the early stage could be an indicator of GS not yet diagnosed. Moreover, Baykul et al., 2003 [15] encouraged oral care professionals to pay close attention to these early signs.

Larsson Wexell et al., 2019 [16] described a case of a woman with a GS diagnosis several years after osteoma radiographic findings, and papers by Alvarez Salgado et al., 2017 [17] and Seehra et al., 2016 [2] noted how the “[…] identification of the dental features of GS during routine orthodontic assessment resulted in the early diagnosis of this condition”.

Another example of the early manifestation of this syndrome throughout osteomas of the jaw is well outlined by Koppany et al., 2002 [18], demonstrating that mandibular bone lumps are among the earliest GS signs. Fotiadis et al., 2005 [19] affirmed that when a GS patient has oral osteomas, they anticipate intestinal polyposis.

A screening method for GS has been proposed and it consists of the finding of three or more cranio-maxillo-facial osteomas [20]. Finally, the work of Herford et al., 2013 [10] depicted the clear correlation between osteomas of the head and this intestinal pathology, reiterating the critical relevance of oral surgeons in early diagnosis.

To the best of our knowledge, all the studies supporting the theory that osteomas can be considered as the first clinical sign of GS, even preceding multiple intestinal polyps, are alphabetically listed in Table A1.

## 5. Conclusions

GS has several intestinal manifestations, which can lead to a negative prognosis if untreated. Adenomatous polyposis is demonstrated to be a predisposing condition toward CRC, and, for this reason, early detection can exert a strong impact on patients’ lives. Among the extra-intestinal signs, disorders in maxillallo-facial district can precede the main abdominal symptoms; in particular, osteomas of the jaw may be an important indicator whose presence could guide dentists and oral surgeons to perform further clinical investigations. In our study, the whole sample presented osteomas of the jaw and/or the facial skeleton, paranasal sinuses included. These results agree with the literature and indicate osteomas of the jaw and the maxillofacial skeleton as a predictive sign of GS. The authors recommend routine panoramic X-ray in teenagers to inspect any osteomas or oral cysts; thus, we may achieve the early detection of GS and the avoidance of serious risks.

## Figures and Tables

**Figure 1 jcm-12-01496-f001:**
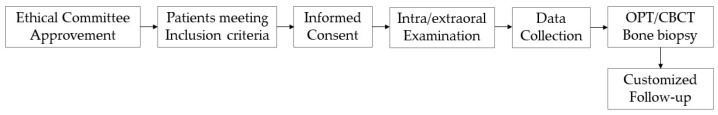
Study methodology.

**Figure 2 jcm-12-01496-f002:**
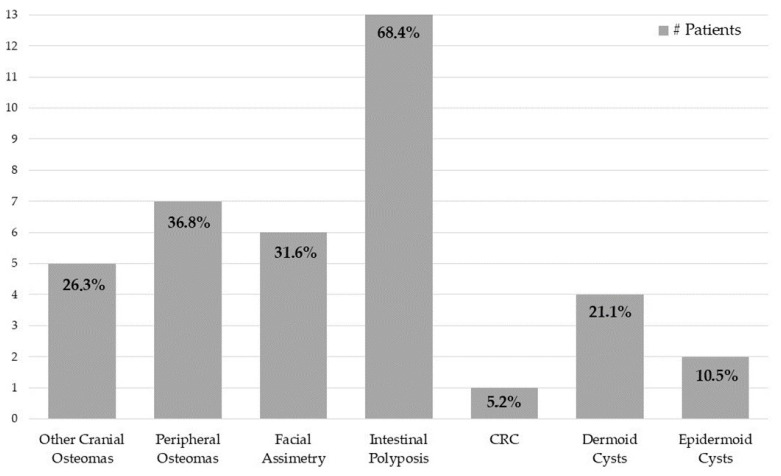
Extraoral characteristics in our GS patients.

**Figure 3 jcm-12-01496-f003:**
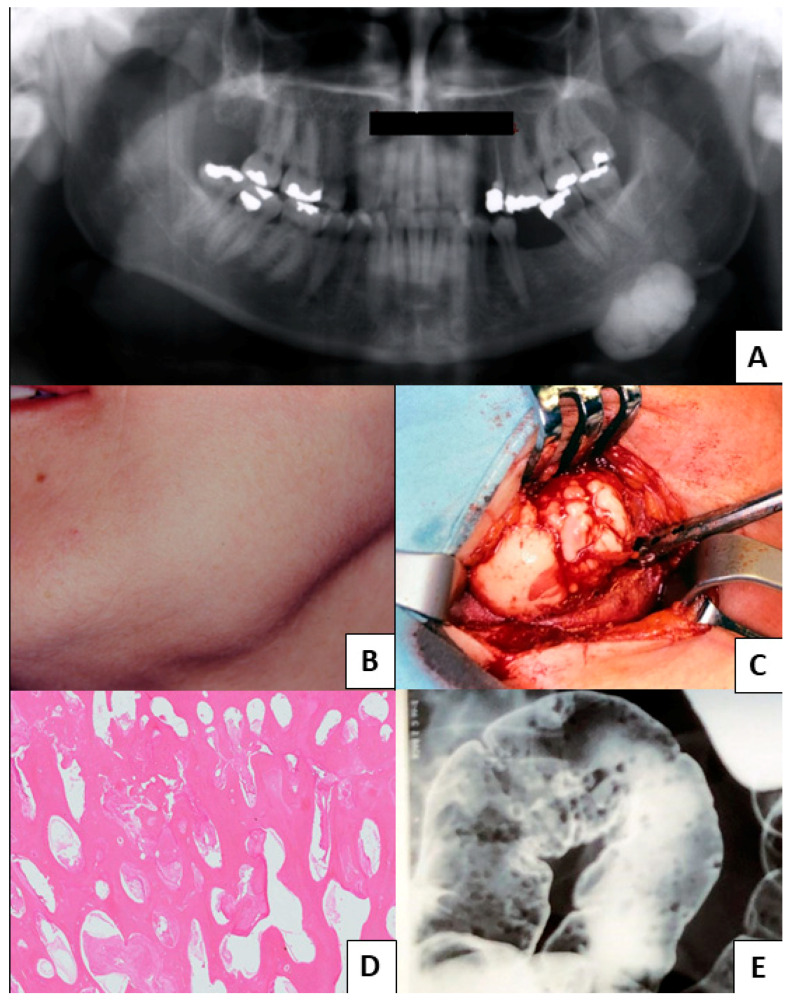
Clinical Case 1. (**A**–**D**) Spongious osteoma of the lower margin of the mandible (panoramic radiogram, facial dysmorphism, intraoperative images, and histological hematoxylin–eosin, respectively); (**E**) the double-contrast barium enema shows the multiple polypoid lesions found in the same patient’s colon.

**Figure 4 jcm-12-01496-f004:**
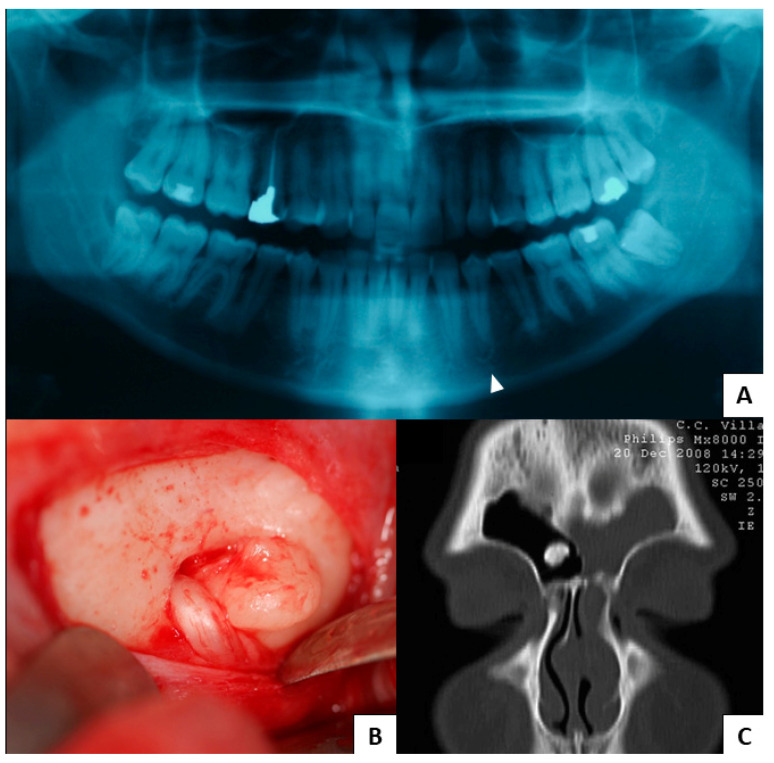
Clinical Case 2. (**A**) A GS patient complaining of pain in lower alveolar nerve region shows an anomaly of the ipsilateral mental foramen in panoramic radiogram (arrow); (**B**) intraoperative image of the osteoma associated with the nerve; (**C**) the same patient also showed an osteoma of the frontal sinus.

**Figure 5 jcm-12-01496-f005:**
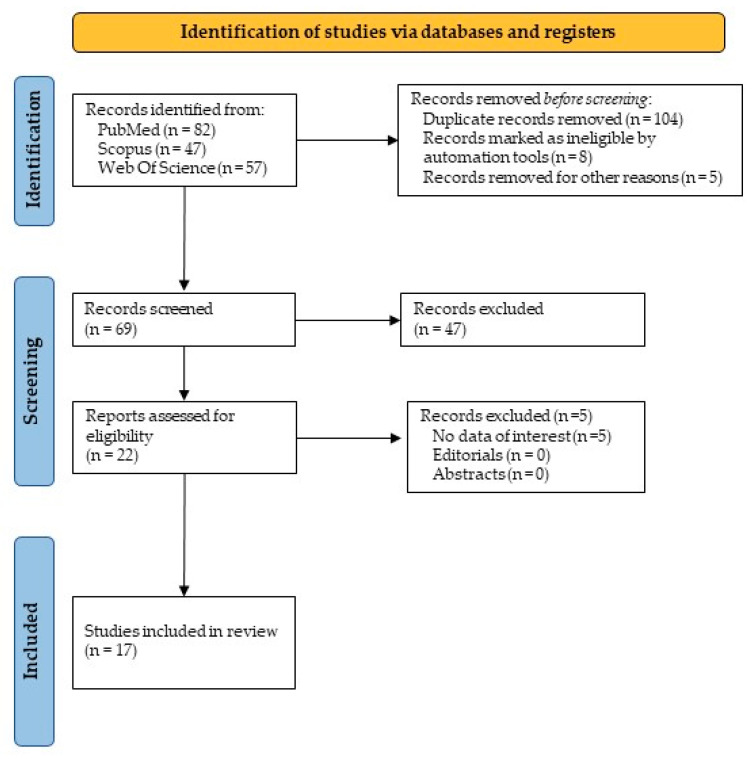
PRISMA flowchart and literature results.

## Data Availability

Not applicable.

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
