# Peer review of "Osteoma of the Jaw as First Clinical Sign of Gardner’s Syndrome: The Experience of Two Italian Centers and Review"

_jcm, 2023, doi:10.3390/jcm12041496_

Round 1

Reviewer 1 Report

It is a well-written work on a well-known and previously documented pathology.

This study does not include novel contributions on the subject, but it is attractive as an update of knowledge.

The aim of the study is to highlight if maxillofacial osteomas could represent an early detection symptom of GS, it is NOT clarified in the series of cases (19) that present.

The proposed systematic review is underdeveloped and could be discussed with the information obtained from its series.

Sorry, I can't find the CRC abbreviations in the text

In short, it is an interesting work as an update on a pathology

Author Response

Reviewer 1

The authors thank Reviewer 1 for the contribution to the improvement of the manuscript. The replies to the comments of Reviewer 1 are below.

  • The aim of the study is to highlight if maxillofacial osteomas could represent an early detection symptom of GS, it is NOT clarified in the series of cases (19) that present.
    • Reply: the authors added the current clarification to the introduction of the manuscript.
  • The proposed systematic review is underdeveloped and could be discussed with the information obtained from its series.
    • Reply: the discussion of the manuscript was modified in line with the suggestion of the reviewer.
  • Sorry, I can't find the CRC abbreviations in the text.
    • Reply: the authors apologize for the inconvenience. The authors added the full-length form of “CRC” to the manuscript.

The authors hope that at the current version of the manuscript is suitable for publication in the Journal of Clinical Medicine.

With warm regards,

The corresponding Author,

Dr. Fabio Dell’Olio

Reviewer 2 Report

The subject of the manuscript is very interesting.

Although the introduction seems too long especially if compared with the discussion section this appear to be useful for the reader to get a broad overview on an unusual topic. The idea of associating the methodology of the literature review process is good.

Possibly it could  be made clearer what is the actual diagnostic pathway in the clinical routine of the two centres in order to specify the study design for clarity of exposition and for the benefit of the reader, see line 130 : “ people who do not have a known FAP history and a positive APC gene test were driven to further gastrointestinal investigation “ 

That said the work  is understandable and it was a stimulating read.

Author Response

Reviewer 2

The authors thank Reviewer 2 for the precious suggestions provided to improve the manuscript. The reply to the comment of Reviewer 2 is below.

  • Possibly it could be made clearer what is the actual diagnostic pathway in the clinical routine of the two centers in order to specify the study design for clarity of exposition and for the benefit of the reader, see line 130: “ people who do not have a known FAP history and a positive APC gene test were driven to further gastrointestinal investigation “ - Patients with a known FAP/CRC family or personal history were included in a recall program for clinical and radiological evaluations, whereas those with a negative history for both FAP/CRC received APC gene testing and further gastrointestinal investigation.
    • Reply: the authors apologize for the inconvenience. The updated version of the methods section reports that patients with a known FAP/CRC family or personal history were included in a recall program for clinical and radiological evaluations, whereas those with a negative history for both FAP/CRC received APC gene testing and further gastrointestinal investigation.

The authors hope that at the current version of the manuscript is suitable for publication in the Journal of Clinical Medicine.

With warm regards,

The corresponding Author,

Dr. Fabio Dell’Olio

Reviewer 3 Report

Osteomas of the jaws as first clinical signs of Gardner’s Syndrome: the experience of two Italian centers and review.

Dear Authors,

Thank you for your effort in writing this manuscript. However, the comments are for better publication quality.

Comments:

Line 28 FAP, Kindly write the full name

Line 31-31, From “GS’s incidence……….16000”, Kindly specify the countries or regions of each incidence category

Line 122, CRC kindly write the full name or CRC be added after colorectal cancer between ()

Results:

-Panoramic views should be included and spotted the osteomas and cysts

-Skeletal X-ray should be pointed out the extraoral osteomas and cysts

-Histopathological figures should be more detailed with more figures

-Figures for facial asymmetry

-You did not mention any dental abnormalities in your results (impaction, supernumerary teeth, hypodontia)

Discussion

Line 190 CHRPE, kindly write the full name or add a list of abbreviations

Line 196,197 did you mean oral osteomas, kindly, this sentence is not clear

Conclusions and recommendations:

Routine panorama for teenagers should be mandatory to inspect any osteomas or  oral cysts, thus, early detection of GS and avoiding hazardous risks.

Author Response

Reviewer 3

The authors are grateful to Reviewer 3 for the precious contribution provided to the improvement of the manuscript. The replies to the comments of Reviewer 3 are below.

  • Line 28 FAP, Kindly write the full name –
    • Reply: the authors kindly highlight that the full name of “FAP” is in line 25 (familial adenomatous polyposis).
  • Line 31-31, From “GS’s incidence……….16000”, Kindly specify the countries or regions of each incidence category
    • Reply: the authors added the required information to the introduction section of the manuscript.
  • Line 122, CRC kindly write the full name or CRC be added after colorectal cancer between ()
    • Reply: the authors apologize for the inconvenience. The updated manuscript has the full name and the abbreviation that were necessary.
  • Results:

-Panoramic views should be included and spotted the osteomas and cysts

-Skeletal X-ray should be pointed out the extraoral osteomas and cysts

-Histopathological figures should be more detailed with more figures

-Figures for facial asymmetry

-You did not mention any dental abnormalities in your results (impaction, supernumerary teeth, hypodontia)

  • Reply: the authors fulfilled those requests of Reviewer 3.
  • Line 190 CHRPE, kindly write the full name or add a list of abbreviations – the full name is in lines 100-101 (congenital hypertrophy of the retinal pigmented epithelium).
    • Reply: the authors apologize for the inconvenience. The updated manuscript has the full name and the abbreviation that were necessary.
  • Line 196,197 did you mean oral osteomas, kindly, this sentence is not clear
    • Reply: the authors added “oral” to the sentence to improve the clarity of the manuscript.
  • Conclusions and recommendations: Routine panorama for teenagers should be mandatory to inspect any osteomas or oral cysts, thus, early detection of GS and avoiding hazardous risks.
    • Reply: the authors thank Reviewer 3 for this precious suggestion. The revised manuscript carries those recommendations in the conclusion section.

The authors hope that at the current version of the manuscript is suitable for publication in the Journal of Clinical Medicine.

With warm regards,

The corresponding Author,

Dr. Fabio Dell’Olio